# Evolution of two metabolic genes involved in nucleotide and amino acid metabolism in *Pseudomonas aeruginosa*

Yutong Wu[1‡], Yuqi Shi[2‡], Xiaohui Liang[3]*

1 Gansu Provincial Hospital of TCM, Gansu University of Chinese Medicine, Lanzhou, Gansu, China, 2 The State Key Laboratory of Pharmaceutical Biotechnology, School of Life Sciences, Nanjing University, Nanjing, Jiangsu, China, 3 Department of Critical Care Medicine, Nanjing Drum Tower Hospital, The Affiliated Hospital of Nanjing University Medical School, Nanjing, Jiangsu, China

‡ These authors share first authorship on this work
* 439397013@qq.com

**Data Availability Statement:** All the detailed information related to the strains is shown in Supplemental tables.

**Funding:** The author(s) received no specific funding for this work.

## Abstract

*Pseudomonas aeruginosa* is an opportunistic human pathogen causing various severe infections. Understanding genetic mechanisms of its metabolic versatility aids in developing novel antibacterial drugs and therapeutic strategies to address multidrug-resistant *P. aeruginosa* infections. The metabolism of nucleotides and amino acids contributes to the cycle of two key biological macromolecules in the genetic central dogma. Guanine deaminase (GuaD) catalyzes the deamination of guanine to produce xanthine to maintain the homeostasis of the nucleotide pool, and transporters specific to BCAAs (termed as BraT) import BCAAs to keep its intracellular availability level. However, little is known about the evolution of GuaD and BraT in *P. aeruginosa* population. Here, two copies turned out to be widespread in *P. aeruginosa* population for each of GuaD and BraT. The phylogenic analysis demonstrated that GuaD1 and BraB were inherited from the ancestor of *Pseudomonas*, while GuaD2 and BraZ were additionally acquired via evolutionary events in the ancestors of *P. aeruginosa*. The functional divergence of two copies was supported by different distribution patterns of dN/dS ratios, divergent expression levels, differentially co-expressed genes, and their functional enrichment modules with few intersections. Besides, some co-expressed genes with known functions are involved in infecting hosts, forming biofilm and resisting antibiotic treatment. Taken together, functional divergence following copy number increase and differentiation of co-expression networks might confer greater metabolic potential to *P. aeruginosa*, especially in response to host immune responses and antibiotic treatments in clinical settings.

## Introduction

*Pseudomonas aeruginosa* is a highly prevalent opportunistic pathogen that causes various types of nosocomial, acute and chronic infections, including ventilator-associated pneumonia [1, 2], sepsis and sepsis shock [3], catheter-associated urinary tract infections [4], and surgical site

**Competing interests:** The authors have declared that no competing interests exist.

infections [5], posing significant challenges to healthcare systems and a huge economic burden on society. *P. aeruginosa* infections could lead to high morbidity and mortality of patients diagnosed with cystic fibrosis [6, 7], cancer [8, 9], burns [10, 11] and immunocompromised symptoms [12, 13]. The World Health Organization (WHO) has recently listed carbapenem-resistant *P. aeruginosa* as one of the most critical threats, necessitating the urgent development of new antibiotics to combat its infection [14].

The transition of *P. aeruginosa* from environmental habitats, such as soil and water, to clinical human-associated settings [15, 16], has presented key challenges for this pathogen to adapt to environmental shifts including clinical antibiotic therapy and immune stress [16, 17]. The metabolic versatility of *P. aeruginosa* plays a crucial role in its pathogenicity and adaptability [18, 19]. Previous studies have identified and characterized over 400 transcription factors that regulate the metabolic state and environmental adaptation of *P. aeruginosa* [20], elucidating the upstream regulatory mechanisms of metabolic alterations. However, there is a paucity of research on the evolution of metabolism-related genes, which are direct participants in metabolic processes and regulatory targets of transcription factors.

Focusing on the evolution of metabolism-related genes, an initial survey revealed that 14 metabolism-related genes exhibited two copies in several typical strains of *P. aeruginosa*, whereas only one copy was found in other investigated species. This finding prompted the selection of two metabolic genes, guanine deaminase (GuaD) and transporters specific to branched-chain amino acids (BCAAs), for further investigation. GuaD catalyzes the hydrolytic deamination of guanine to produce xanthine, which is an important component of the guanine degradation pathway in bacteria. Transporters specific to BCAAs (termed as BraT) import BCAAs to keep its intracellular availability level, involed in some key biological processes. The metabolic homeostasis of nucleotides and amino acids is crucial for the cycle of two key biological macromolecules in the genetic central dogma. Therefore, these two metabolic genes could contribute to *P. aeruginosa*'s clinical adaptation due to the functional diversity of two copies. This study intends to address the following questions through phylogenetic, comparative genomics and transcriptional analyses: (1) Is the copy number increase common at the population level in *P. aeruginosa*? (2) What is the evolutionary origin of the two copies? (3) Have the biological functions of two copies diverged? (4) How does the coexistence of two copies contribute to the adaptation of *P. aeruginosa* to clinical settings? By answering these questions, this study will enhance our understanding of the evolution and functional implications of metabolism-related genes in *P. aeruginosa*, particularly in the context of clinical adaptation.

## Materials and methods

### Bacterial strains

Genomic sequences of 1945 strains from 565 species in γ-Proteobacteria were downloaded from the NCBI reference database (https://www.ncbi.nlm.nih.gov/refseq/). Detailed information related to the strains is shown in S1 Table. In addition, the metadata of 391 *P. aeruginosa* strains was collected, and the descriptions related to habitat were extracted, including host, isolation source, clinical isolates and environmental information, as detailed in S2 Table.

### Phylogeny of strains

The phylogeny of strains was constructed using the concatenation of the 120 ubiquitous marker genes proposed by previous studies [21]. In brief, HMM files of each gene were searched and downloaded within Pfam v27 and TIGRFAMs v15.0 databases. The 120 marker proteins were identified from each strain genome using a HMM search implemented in

HMMER v3 with e-value = $1e^{-20}$ [22]. The obtained proteins were firstly concatenated using Faops (https://github.com/wang-q/faops), followed by multiple sequence alignment with MUSCLE v3 [23] and the resulting sequences were trimmed using trimAl v1.5.0 with default parameters [24]. The phylogenetic tree was inferred using IQ-TREE v2 with 1000 bootstrap replicates [25]. iTOL v5 (Interactive Tree of Life) was used to visualize and annotate the tree [26].

### Identification of protein sequences of two metabolic genes

To avoid model bias, the HMM files of two metabolic genes were downloaded within three databases, including TIGR00796 and TIGR02967 in TIGRFAMs v15.0, PF05525 in Pfam v35.0, and PTHR30588 and PTHR11271:SF6 in PATHER v16.0. Using the HMM files as query sequences, the candidate protein sequences of two metabolic genes were identified in each genome using a HMM search implemented in HMMER v3 with e-value = $1e^{-20}$ [22]. All hits were further analyzed with hmmscan in HMMER v3 against three local databases, TIGR-FAMs v15.0, Pfam v35.0 and PATHER v16.0, with e-value = $1e^{-50}$. When two or more domains were annotated for a single gene, the domain with the lowest e-value was retained. The resulting protein sequences then were used to run three rounds of BLASTP (v2.10.0) analysis against each strain genome with e-value = $1e^{-20}$.

### Phylogenetic relationships of protein sequences of two metabolic genes

Sequence alignment was performed using mafft v7 with default parameters [27]. The aligned results were trimmed by trimAl v1.5.0 [24] with default parameters. Phylogenetic relationships were constructed using IQ-TREE v2 with 1000 bootstrap replicates [25], and then visualized and annotated with iTOL v5 [26].

### Genomic analysis of two metabolic genes

Genomic regions flanking two metabolic genes were compared using Clinker [28], and *P. aeruginosa* PAO1 was used as a reference. The pangenome landscape of 391 *P. aeruginosa* strains was analyzed using PPanGGOLiN v1.0 [29]. Gephi v0.10.1 was used to visualize and annotate the resulting file [30].

### Evolutionary estimation of two metabolic genes

Pairwise sequence divergence, including dN, dS and dN/dS, was calculated using MEGA X with Nei-Gojobori model [31]. Six species with more than 30 strains were included, namely *P. aeruginosa*, *P. chlororaphis*, *P. syringae*, *P. fluorescens*, *P. putida* and *P. stutzeri*. The frequency of dN/dS was visualized using the ggplot2 package (v3.2.2) [32].

### Functional analysis of two copies of two metabolic genes

Transcriptional datasets of *P. aeruginosa* strains were downloaded from the NCBI GEO database. Treatment conditions and sample information are available in S3 Table. Normalization of raw transcriptional data was performed with RMA function in the affy package [33]. 203 samples were further split into 40 groups for subsequent analysis. Detailed grouping information is available in S3 Table. A total of 64000 combinations were obtained after performing a three-three combination with 40 groups. The WGCNA R-package was used to identify the respective co-expressed genes of two metabolic genes in the 64000 combinations [34]. The most significant co-expressed genes were determined using the 3σ principle. A venn diagram of co-expressed genes was visualized using the ggplot2 package (v3.2.2) [32]. Clustering

analysis was performed based on the mean of normalized transcript data in 40 groups using the pheatmap package [35]. Functional categorizations of co-expressed genes were annotated using eggNOG-Mapper v4.5 [36]. The GO pathway enrichment was identified using the enricher function implemented in the clusterProfiler package (v3.12.0) with cutoff criteria of p-values < 0.01 and FDR < 0.05, aiming to determine their biological significance. The results were visualized with the ggplot2 package (v3.2.2) [32].

## Results

### Widespread existence of two copies of two metabolic genes in *P. aeruginosa*

The phylogeny of 391 *P. aeruginosa* strains was constructed using the concatenation of the 120 marker proteins, and three representative species from *Pseudomonas* were used as outgroups, including *P. fluorecens*, *P. syringae* and *P. putida*. The phylogenetic tree revealed three distinct clades with representative strains of PA7, UCBPP_PA14, and PAO1 (S1 Fig), as previously defined by Freschi et al. [37]. A total of 779 guanine deaminases (GuaD) were identified in 391 *P. aeruginosa* strains, with two copies in each of 378 strains and one copy in each of the remaining three strains (S1 Fig and S4 Table). Phylogenetic analysis showed that guanine deaminases in *P. aeruginosa* formed two distinct clades, named as GuaD1 clade containing GuaD1 in PAO1 and GuaD2 clade containing GuaD2 in PAO1, respectively (Fig 1A). A total of 773 transporters specific to BCAAs (BraT) were identified from the genomes of 391 *P. aeruginosa* strains, with two copies in each of 382 strains and one copy in each of the remaining nine strains (S1 Fig and S5 Table). Phylogenetic analysis showed that transporters specific to BCAAs in *P. aeruginosa* formed two distinct clades, named as Clade I containing BraB in PAO1 and Clade II containing BraZ in PAO1, respectively (Fig 1A). Besides, Clade II was further clustered into two sub-clades, named as sub-clade1 and sub-clade2, respectively. 22 strains exhibited different situations from that of *P. aerugonisa* PAO1. Detailed information was available in S2–S4 Figs.

Genomic analysis of four commonly used reference strains, namely PAO1, LESBS8, UCBPP_PA14 and PA7, showed that they shared the same genetic organization of genomic regions flanking the two copies of the two metabolic genes, except that only the upstream of *guaD1* in LESB58 demonstrated good co-linearity with that of *guaD1* in PAO1 (Fig 1B). However, the genomic regions flanking two copies of each gene differed from each other, with some similarity only between two copies, either for *guaD* or *braT* (S5 Fig). Furthermore, pangenome analysis of 391 *P. aerugonisa* strains further revealed that both genomic organizations of each copy about two metabolic genes belonged to persistent genome, with more than five genes either upstream or downstream maintaining stable (Fig 1C). Notably, the genomic regions flanking *guaD1* exhibited two different gene arrangements, with one resembling *guaD1* in PAO1 and the other according with *guaD1* in LESB58 (S6 Fig).

### Evolution of two metabolic genes in *Pseudomonas*

Of 49 *Pseudomonas* species, only *P. aeruginosa* was found to possess two copies, either guanine deaminase or transporters specific to BCAAs (Table 1 and S4 and S5 Tables). The phylogeny of 49 *Pseudomonas* species demonstrated four apparent groups (Fig 2A left), including *P. aeruginosa* group, *P. putida* group, *P. syringe* group and *P. fluorescens* group, consistent with previous studies [38]. However, comparative analysis showed topological inconsistency between species tree and protein tree, both for guanine deaminase and transporters specific to BCAAs (Fig 2A center and right). The protein tree of guanine deaminase demonstrated that GuaD1 in *P. aeruginosa* clustered with guanine deaminases (GuaD) in other *Pseudomonas* species, further forming a sister clade to GuaD2 in *P. aeruginosa* (Fig 2A center). The phylogenetic tree of

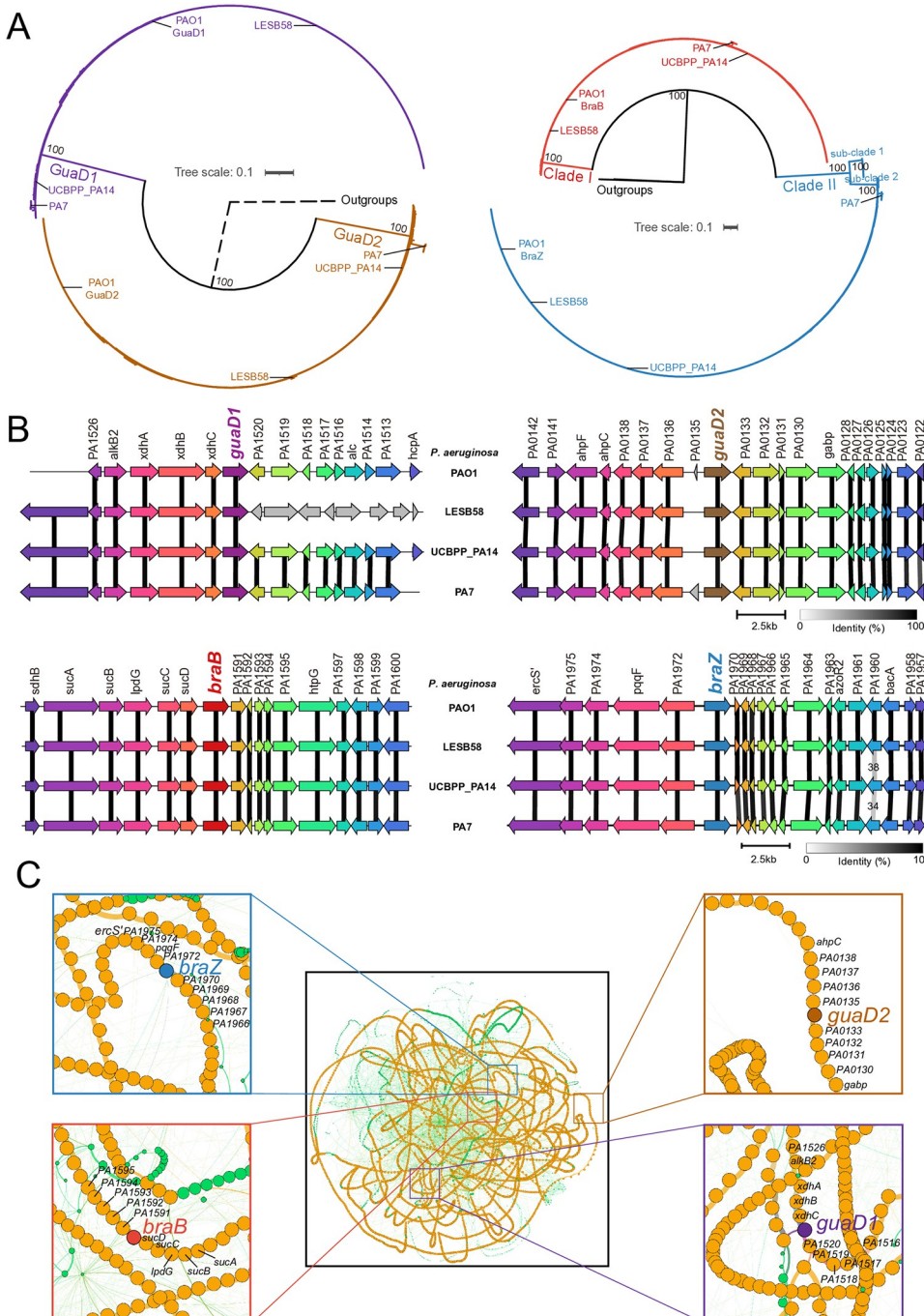

**Fig 1. Widespread existence of two copies in *P. aeruginosa*.** **(A)** The phylogenetic relationships of guanine deaminases (left) or transporters specific to BCAAs (right) in 391 *P. aeruginosa* strains. The scale bar represents 0.1 substitutions per site and the bootstrap values of major clades are shown at the nodes. The dashed branches indicate the long branches linking the outgroups to the remainder of the tree. The clade consisting of GuaD1 was named as GuaD1 clade and colored with purple. And the clade consisting of GuaD2 was named as GuaD2 clade and colored with brown. The clade consisting of BraB was named as Clade I and colored with red. And the clade consisting of BraZ was named as Clade II and colored with blue. Clade II was further clustered into two sub-clades, named as sub-clade1 and sub-clade2, respectively. Two copies of two metabolic genes in four representative strains are indicated, respectively. **(B)** Comparative analysis of genomic regions flanking guanine deaminase (top) or transporters specific to BCAAs (bottom) in four representative strains of *P. aeruginosa* referred to PAO1. Colorful arrows and dark shading indicate the gene direction and nucleotide sequence identity of conserved regions. The nucleotide sequence identity less than 90% is indicated. The scale bar indicates the length of 2.5kb nucleotides. **(C)** Pangenome graph of 391 *P.*

*aeruginosa* genomes. Edges correspond to genomic colocalization and nodes correspond to gene families. The thickness of the edges is proportional to the number of genomes sharing that link. The size of the nodes is proportional to the total number of genes in each family. The edges between persistent, shell and cloud nodes are colored in orange, green and blue, respectively. Nodes are colored in the same way. The frames show a zoom of genomic region containing *guaD1* and its flanking genes, genomic region containing *guaD2* and its flanking genes, genomic region containing *braB* and its flanking genes, and genomic region with *braZ* and its flanking genes, respectively. Besides, *guaD1*, *guaD2*, *braB* and *braZ* are highlighted by purple, brown, red and blue, respectively.

transporters specific to BCAAs (BraT) in *Pseudomonas* demonstrated that two distinct clades were identified and named as Clade I and Clade II, consisting of BraB and BraZ, respectively (Fig 2A right).

For guanine deaminase, nine *Pseudomonas* species were included to explore the stability of genes flanking *guaD*, excluding *P. alcaligenes* due to the absence of guanine deaminase.

**Table 1. Copy number of two metabolic genes in each strain from γ-proteobacteria species.**

| Taxonomy | Number of assemblies | Number of guanine deaminase | Average per genome | Number of transporters specific to BCAA | Average per genome |
|---|---|---|---|---|---|
| *P. aeruginosa* | 391 | 779 | 2 | 773 | 2 |
| *P. chlororaphis* | 59 | 59 | 1 | 59 | 1 |
| *P. entomophila* | 3 | 3 | 1 | 3 | 1 |
| *P. fluorescens* | 38 | 37 | 1.2 | 38 | 1 |
| *P. protegens* | 23 | 23 | 1 | 23 | 1 |
| *P. putida* | 48 | 49 | 1 | 49 | 1 |
| *P. savastanoi* | 5 | 5 | 1 | 5 | 1 |
| *P. stutzeri* | 30 | 30 | 1 | 30 | 1 |
| *P. syringae* | 40 | 40 | 1 | 40 | 1 |
| Other *Pseudomonas* spp. | 190 | 184 | 1 | 190 | 1 |
| Aeromonadaceae | 17 | 9 | 0.5 | 25 | 1.5 |
| Bruguierivoracaceae | 1 | 1 | 1 | 1 | 1 |
| Budviciaceae | 4 | 0 | 0 | 8 | 2 |
| Cardiobacteriaceae | 2 | 0 | 0 | 2 | 1 |
| Enterobacteriaceae | 71 | 40 | 0.6 | 75 | 1.1 |
| Erwiniaceae | 18 | 16 | 1 | 18 | 1 |
| Hafniaceae | 6 | 7 | 1.1 | 13 | 2.2 |
| Legionellaceae | 16 | 1 | 0.1 | 7 | 0.4 |
| Moraxellaceae | 523 | 486 | 0.9 | 488 | 0.9 |
| Morganellaceae | 19 | 1 | 0.1 | 25 | 1.3 |
| Orbaceae | 3 | 0 | 0 | 2 | 0.7 |
| Pasteurellaceae | 34 | 0 | 0 | 41 | 1.2 |
| Pectobacteriaceae | 28 | 13 | 0.5 | 28 | 1 |
| Pseudoalteromonadaceae | 33 | 6 | 0.2 | 2 | 0.1 |
| **Pseudomonadaceae** | 13 | 6 | 0.4 | 7 | 0.5 |
| Shewanellaceae | 52 | 0 | 0 | 73 | 1.4 |
| Vibrionaceae | 46 | 27 | 0.6 | 72 | 1.6 |
| Yersiniaceae | 32 | 19 | 0.6 | 37 | 1.2 |
| Other families | 200 | 74 | 0.4 | 0 | 0 |

Note: (1) Other *Pseudomonas* spp. represent species from the genus *Pseudomonas* other than those listed in Table 1; (2) The number of strains from each family was determined with reference to the evolutionary relationship with the family Pseudomonadaceae consisting of *P. aeruginosa*; (3) The bolded **Pseudomonadaceae** represents the investigated species belonging to Pseudomonadaceae other than *Pseudomonas* species.

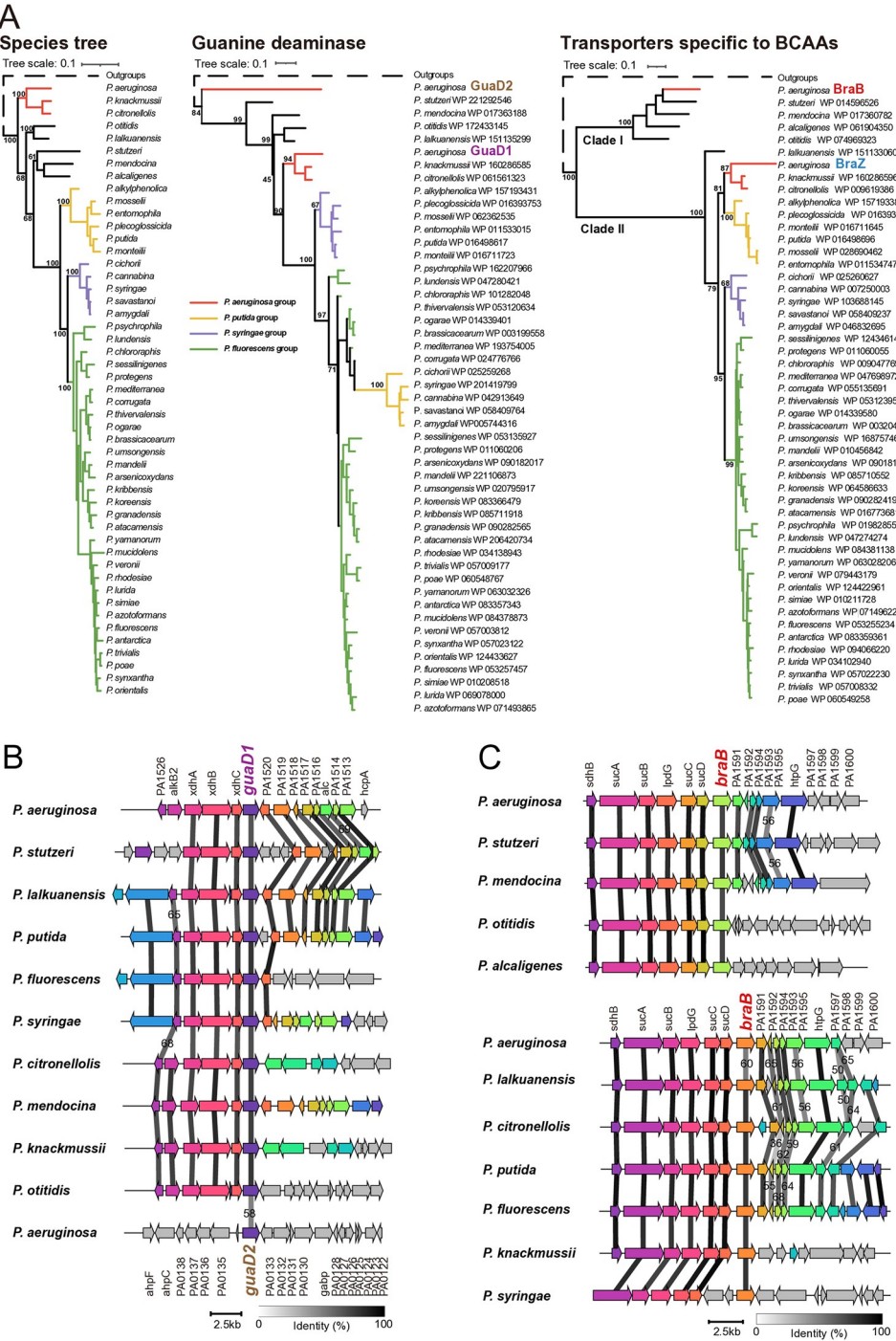

**Fig 2. Phylogeny of two metabolic genes in *Pseudomonas*. (A)** The species tree (left), the phylogenetic tree of guanine deaminases (center) and the phylogenetic tree of transporters specific to BCAA (right) of 49 *Pseudomonas* species. The scale bar represents 0.1 substitutions per site and the bootstrap values of major clades are shown at the nodes. Four assigned groups were marked with colorful branches. Red, orange, purple and green branches represent *P. aeruginosa* group, *P. putida* group, *P. syringae* group and *P. fluorescens* group, respectively. The dashed branches indicate the long branches linking the outgroups to the remainder of the tree. Transporters specific to BCAAs formed two clades named as Clade I and Clade II, respectively. **(B)** Comparative analysis of genomic regions flanking guanine deaminases in *Pseudomonas* referred to *P. aeruginosa* PAO1. Colorful arrows and dark shading indicate the gene direction and nucleotide sequence identity of conserved regions. Nine species in *Pseudomonas* compared to genomic regions flanking *guaD1* and *guaD2* in *P. aeruginosa* PAO1. The nucleotide sequence identity less than 70% are

indicated. The scale bar indicates the length of 2.5kb nucleotides. **(C)** Comparative analysis of genomic regions flanking transporters specific to BCAAs in *Pseudomonas* referred to *P. aeruginosa* PAO1. Colorful arrows and dark shading indicate the gene direction and nucleotide sequence identity of conserved regions. Top, species in the Clade I compared to genomic regions flanking *braB* in PAO1. Bottom, species in the Clade II compared to genomic regions flanking *braB* in PAO1. The nucleotide sequence identity less than 70% are indicated. The scale bar indicates the length of 2.5kb nucleotides.

Comparative analysis demonstrated that nine species shared the same gene clusters with *P. aeruginosa* PAO1, either the upstream or genomic region flanking *guaD*. However, no similarity was found between genomic region flanking *guaD* in each of nine species and that of *guaD2* in *P. aeruginosa* PAO1 (Fig 2B). For transporters specific to BCAAs, genomic analysis of five species in Clade I showed that two species shared the same genetic organization of genomic regions flanking *braB* in *P. aeruginosa* PAO1, including *P. stuzeri* and *P. mendocina*, and the remaining two species had the same gene arrangement with the upstream of *braB* (Fig 2C top). For Clade II, the representative species of three groups, as well as all remaining species excluding three groups, were subjected to comparative genomic analysis. Surprisingly, there was no co-linearity between each of all the investigated species and *P. aeruginosa* PAO1 regarding the genomic regions flanking *braZ* (S7 Fig). Further comparative analysis found that four species shared the same genetic organization of genomic regions flanking *braB* in *P. aeruginosa* PAO1, including *P. lalkuanensis*, *P. citronellolis*, *P. putida* and *P. fluorescens*, and the remaining two species had a similar gene arrangement with the upstream of *braB* (Fig 2C bottom). Besides, 101 kb of genomic region flanking *braB* or *braZ* was used to blast against other species in *Pseudomonas*. In contrast to *braB*, about 9kb of genomic region flanking *braZ* might be obtained in the ancestors of *P. aeruginosa*, with absent in other species (S8 Fig). In summary, GuaD1 and BraB in *P. aeruginosa* might have been inherited from the ancestor of *Pseudomonas*, while GuaD2 and BraZ were additionally acquired through evolutionary events in the ancestors of *P. aeruginosa*.

## Evolutionary source of two metabolic genes

To further explore the evolutionary source of GuaD2 and BraZ in *P. aeruginosa*, 1945 genomes of 565 species from γ-Proteobacteria were analyzed. 249 species with guanine deaminase were subject to the phylogenetic analysis. As shown in Fig 3A, a phylogenetic signal indicative of HGT event was found that guanine deaminase (GuaD) from a non-*Pseudomonas* species (*Cellvibrio japonicus*) clustered with GuaD2. Comparative analysis showed that genomic region flanking *guaD* in *C. japonicus* possessed homologous genes with the upstream of *guaD2* (Fig 3B left), as well as the upstream and downstream of *guaD1* (Fig 3B right) in *P. aeruginosa* PAO1. To further verify the evolutionary source of GuaD2 in *P. aeruginosa*, three species from related families of *P. aeruginosa* were included (Fig 3A). Comparative analysis of genomic regions flanking *guaD* revealed the presence of conserved gene arrangement (Fig 3C), while three genes were also found in *Thalassolituus oleivorans* for three homologous genes between genomic region flanking *guaD* in *C. japonicus* and that of *guaD2* in *P. aeruginosa* PAO1. Phylogenetic relationships of species and gene arrangement flanking *guaD* supported that *guaD1* and *guaD* in those three species belonged to the orthologous genes, and that *guaD2* was acquired from *C. japonicus* via the HGT event (Fig 3D).

For transporters specific to BCAAs, 383 species were subject to the phylogenetic analysis. As shown in S9 Fig, no phylogenetic signal indicative of HGT event was found that transporters specific to BCAAs from non-*Pseudomonas* species clustered together with BraZ in the phylogenetic tree (S9 Fig). Therefore, the non-redundant protein database (NR) of bacteria was

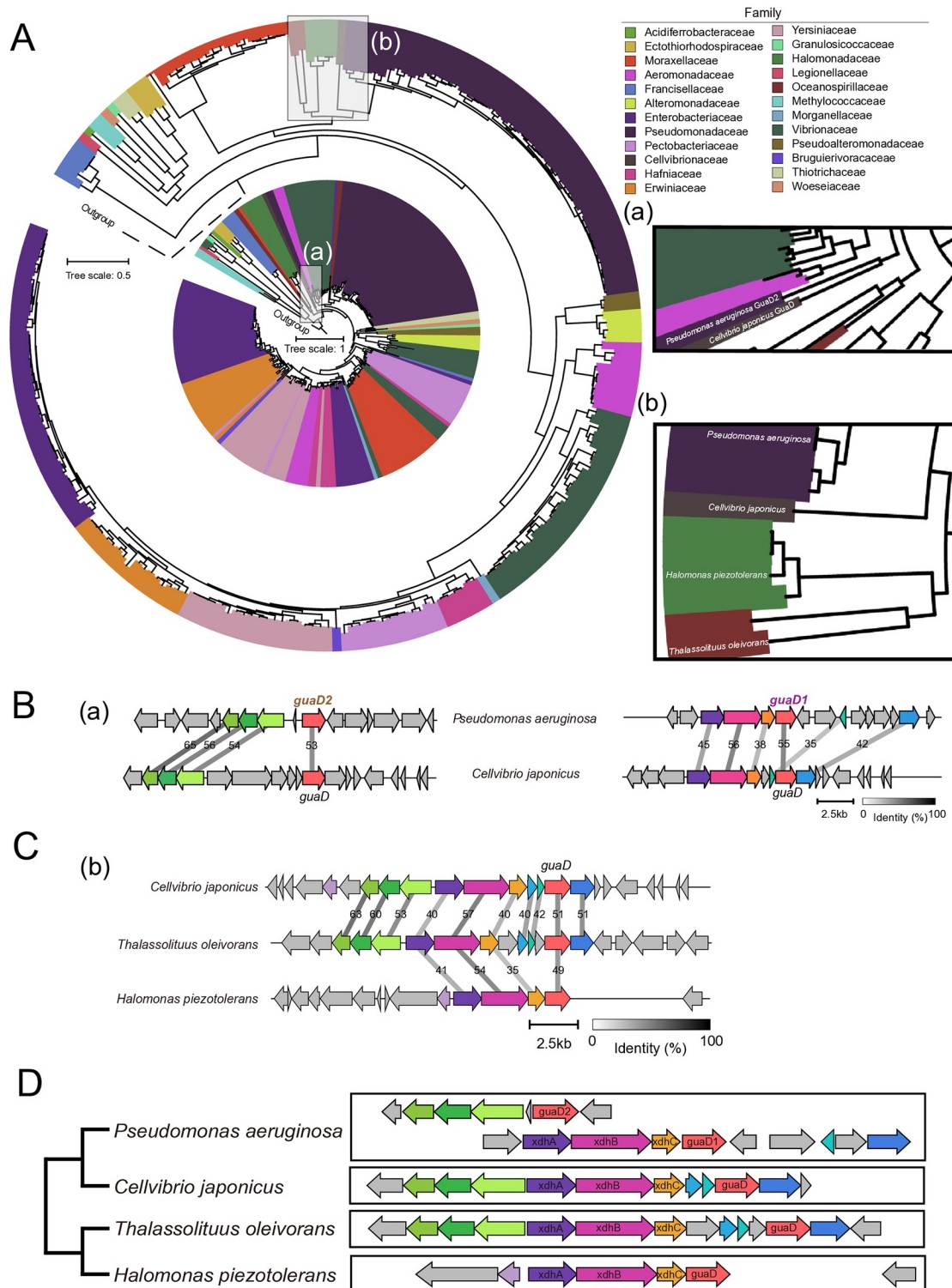

**Fig 3. Evolutionary source of guaD2 in *P. aeruginosa*.** **(A)** The species tree (outside) and the phylogenetic tree of guanine deaminase (inside) of 249 γ-proteobacteria species. The scale bar represents 0.5 substitution per site for the species tree, as well as 1 substitution per site for the protein tree. Species (a) adjacent to *P. aeruginosa* GuaD2 are displayed in a zoom, and species (b) from three related families of *P. aeruginosa* are displayed in a zoom. **(B)** Comparative analysis of genomic regions flanking guanine deaminase in *C. japonicus* referred to *guaD2* (left) and *guaD1* (right) in *P. aeruginosa* PAO1. Colorful arrows and dark shading indicate the gene direction and nucleotide sequence identity of conserved regions. The nucleotide sequence identity less than 70%

are indicated. The scale bar indicates the length of 2.5kb nucleotides. **(C)** Comparative analysis of genomic regions flanking guanine deaminase in two families adjacent to *C. japonicus* referred to *guaD* in *C. japonicus*. Colorful arrows and dark shading indicate the gene direction and nucleotide sequence identity of conserved regions. The nucleotide sequence identity less than 70% are indicated. The scale bar indicates the length of 2.5kb nucleotides. **(D)** The topology of phylogenetic relationships (left) and genomic regions flanking *guaD* (right) of *P. aeruginosa* and three species in related families. The topology without the branch length was extracted from the species tree in Fig 3A.

downloaded from the NCBI database and further used to explore the evolutionary source of BraZ in *P. aeruginosa*. The protein sequences of BraB and BraZ in *P. aerugonisa* PAO1 were used to perform a BLAST search against the NR database, with an identity > 30% and e-value = $1e^{-5}$. All hits were clustered by CD-HIT v4.6 [39], with a threshold of 90% sequence identity after removing duplicate sequences (S10A Fig). Then, the protein sequences in the branches where BraB and BraZ are located, as well as in the neighboring branches, are extracted according to the sequence names. Sequence alignment was performed using mafft with default parameters [27]. The aligned results were trimmed by trimAl v1.5.0 [24] with default parameters. Phylogenetic relationships were constructed using IQ-TREE v2 with 1000 bootstrap replicates [25], and then visualized and annotated with iTOL v5 [26]. The phylogenetic tree showed that transporters specific to BCAAs from non-*Pseudomonas* species clustered together with BraZ, namely *Kerstersia gyiorum*, representing a possible HGT signal (S10B Fig). Further comparative analysis revealed that there was no collinearity between genomic region flanking *braT* in *K. gyiorum* and that of *braZ* in *P. aeruginosa* PAO1 (S10C Fig).

## Functional divergences of two copies about two metabolic genes in *P. aeruginosa*

Frequency distribution of dN/dS values in six *Pseudomonas* species supported that two metabolic genes might experience the purifying selection, as indicated by dN/dS values less than one (Fig 4A). By contrast, the inherited copy showed a similar frequency distribution to that of other *Pseudomonas* species, while the additionally generated copy exhibited different frequency distributions with notably deviating from zero (Fig 4A), implying that respective two copies of two metabolic genes experienced the divergent evolutionary paths. At the transcriptional level, expression patterns of two copies differed from each other, either for *guaD* or *braT* (Fig 4B). The respective co-expressed genes of two copies were identified through the WGCNA procedure, with 78 for *guaD1* and 77 for *guaD2*, as well as 21 for *braB* and 29 for *braZ*. Importantly, there was no intersection between co-expressed genes of two copies (Fig 4C and S11 Fig), indicating possible functional divergence. Pangenome analysis further revealed that all the co-expressed genes belonged to the persistent genome, present in no less than 90% of 391 *P. aeruginosa* strains (S6 Table). Transcriptionally clustering analysis revealed that expression profiles of respective co-expressed genes about two copies differed from each other, except for 4 genes that exhibited special clustering patterns, including PA1205, PA1264, PA2745 and PA5434 (S12 Fig). Furthermore, GO enrichment analysis showed that significantly enriched GO terms were different for co-expressed genes of *guaD1* and *guaD2* (Fig 4D), and that only the top two most enriched GO terms were similar between respective co-expressed genes of *braB* and *braZ*, namely plasma membrane and cell periphery, with large differences in the remaining GO terms (Fig 4E).

## Discussion

*P. aeruginosa* is an opportunistic human pathogen causing a wide array of life-threatening infections [40]. Currently, developing novel antibiotics by adding modifications to extant

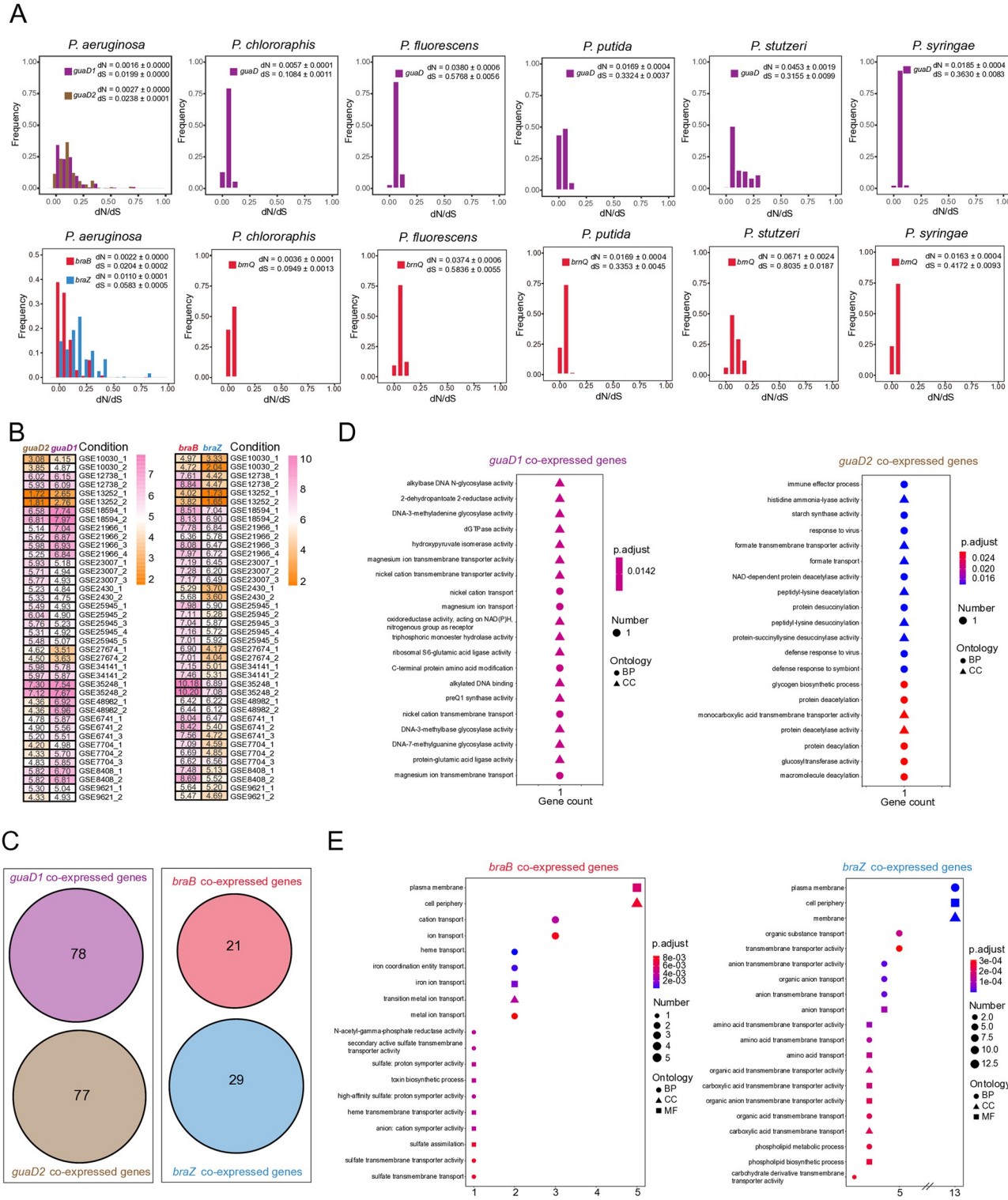

**Fig 4. Functional divergence of two copies about two metabolic genes in *P. aeruginosa*. (A)** The frequency distributions of dN/dS values of guanine deaminases (top) and transporters specific to BCAAs (bottom) from six species in *Pseudomonas*. Six species with more than 30 strains were included, namely *P. aeruginosa*, *P. chlororaphis*, *P. putida*, *P. syringae*, *P. fluorescens* and *P. stutzeri*. Frequency columns of dN/dS values of *guaD1* and *guaD2* in *P. aeruginosa* were colored with purple and brown, respectively. Frequency columns of that of guanine deaminase in other species were colored with the same red color because of orthologous to *guaD1*. Frequency columns of dN/dS values of *braB* and *braZ* in *P. aeruginosa* were colored with red and blue,

respectively. Frequency columns of that of transporters specific to BCAAs in other species were colored with the same red color because of orthologous to *braB*. Both dN and dS were indicated with mean ± SD. **(B)** Expression of *guaD1* and *guaD2* (left), as well as that of *braB* and *braZ* (right) in each of 40 treatment conditions. **(C)** Venn diagram of respective co-expressed genes of *guaD1* and *guaD2* (left), as well as that of *braB* and *braZ* (right). **(D)** GO term enrichment of co-expressed genes of *guaD1* (left) and *guaD2* (right). BP stands for biological process, CC indicates cellular component, and MF represents molecular function. **(E)** GO term enrichment of co-expressed genes of *braB* (left) and *braZ* (right). BP stands for biological process, CC indicates cellular component, and MF represents molecular function.

antibiotics has encountered the bottleneck [41]. Studying molecular mechanisms of metabolic versatility helps the development of novel antibacterial drugs and therapeutic strategies to deal with the multidrug-resistant *P. aeruginosa* infections.

Here, widespread existence of two copies was uncovered in *P. aeruginosa* population, both for guanine deaminase (GuaD) and transporters specific to BCAAs (BraT) involved in nucleotide and amino acid metabolism, respectively. Furthermore, GuaD1 and BraB were inherited from the ancestor of *Pseudomonas*, and GuaD2 and BraZ were additionally acquired in the ancestors of *P. aeruginosa*. Evolutionary analysis revealed that GuaD2 in *P. aeruginosa* was acquired from *C. japonicus* via the HGT event. *C. japonicus* belongs to a terrestrial saprophytic bacterium [42], and the ancestors of *P. aeruginosa* lived mainly in the water and soil environment [15, 16]. The overlap in ecological habitats might have facilitated the occurrence of the HGT event via the direct contact between the "donor" and "recipient". Surprisingly, genomic region flanking *guaD* in *C. japonicus* showed homology to both genomic regions flanking *guaD1* and the upstream region of *guaD2* in *P. aeruginosa* PAO1(Fig 3B). Previous functional studies have shown that it belongs to the key first step of guanine metabolism that GuaD catalyzes the deamination of guanine to produce xanthine, followed by that XdhA and XdhB dimers catalyze the dehydrogenation of xanthine to produce uric acid with help of the cofactor XdhC [43, 44]. For *P. aeruginosa*, *xdhA*, *xdhB* and *xdhC* are closely arranged in the upstream region of *guaD1*, and for three closely related species, *C. japonicus*, *T. oleivorans* and *H. piezotolerans*, only one copy of GuaD was identified with an upstream gene arrangement similar to that of *guaD1* in *P. aeruginosa* PAO1 (Fig 3D), suggesting that GuaD1 and GuaD should belong to orthologous genes and be both inherited from a common ancestor. After the HGT event, the ancestor of *P. aeruginosa* acquired the genomic region containing *guaD* and its eight upstream genes from *C. japonicus*, and subsequently lost its five adjacent genes due to functional redundancy, including evolutionarily conserved *xdhA*, *xdhB* and *xdhC* (Fig 3).

As to the evolutionary source of BraZ, the obtained evidence supports two possible hypotheses: (1) *braZ* might be the result of *braB* duplication, followed by a transposition event; (2) *braZ* was got via the HGT event. For the former, comparative analysis of *Pseudomonas* found that BraZ clustered together with BraT of other species from *P. aeruginosa* group, but not BraB (Fig 2A). BraZ via duplication event might keep a similar evolutionary path with that of closely related species, whereas BraB might experience a distinct evolutionary path. In addition, different basic functions of BraB and BraZ coincide with the fact that one keeps the original functions, and the other tends to acquire new functions, when two copies were generated by gene duplication [45]. The evolutionary divergence was further supported by the significantly different estimations of dN, dS and dN/dS (Fig 4A). However, no genomic traces of the duplication event were left in *P. aerugonisa*. Genomic analysis found that there was no collinearity between genomic regions flanking *braB* and *braZ* (S2B Fig). As to the hypothesis (2), the evolutionary acquisition of BraZ may result from the HGT event. The evidence supporting the above hypothesis consists of two aspects. One is that about 9 kb of genomic region flanking *braZ* in *P. aeruginosa* PAO1 is absent in other *Pseudomonas* species, whereas the genomic region flanking *braB* remains relatively stable in *Pseudomonas* (S8 Fig). Thus, it was proposed that

*braB* was inherited from the ancestor of *Pseudomonas*, and about 9 kb region flanking *braZ* was obtained via the HGT event. The other is about selection patterns of BraT at the population level. The distribution of dN/dS estimations of *braB* is similar to that of other *Pseudomonas* species, whereas *braZ* shows different distribution patterns (Fig 4A), implying functional divergence after evolutionary acquisition. Taken together, the above evidence does not firmly affirm one hypothesis and exclude the other. Given that evolutionary acquisition of BraZ occurred in the ancestors of *P. aeruginosa* inhabiting in soil or water environments [15, 17], exploring the evolutionary source of BraZ might require the inclusion of metagenomic sequencing data of samples associated with the original habitats.

Guanine deaminase (GuaD) catalyzes the hydrolytic deamination of guanine to produce xanthine, which is an important component of the guanine degradation pathway in bacteria [46–48]. For *Pseudomonas*, two copies of guanine deaminases, GuaD1 and GuaD2, were identified in *P. aeruginosa* in this study, whereas only one copy, an ortholog of GuaD1, was present in other *Pseudomonas* species. Recent studies have shown that guanine deaminase accepts only guanine as a substrate and does not catalyze any other base, including adenine and even in the case of bases with minimal substitutions on the guanine scaffold, such as 9-methylguanine and 1-methylguanine [48]. Therefore, no evidence in terms of substrate preference has been obtained to support that the coexistence of two copies in *P. aeruginosa* could be a result of interaction between *P. aeruginosa* and the environment rather than the functional redundancy. In addition, previous studies demonstrated that the expression of *guaD* in bacteria was tightly regulated [49, 50], mainly due to that the mutagenic base xanthine via the hydrolytic deamination of guanine could potentially get incorporated into DNA and RNA, causing genetic aberrations [51]. Here, we found that the expression profiles of two copies differed from each other across the 40 treatment conditions, and that the expression of *guaD1* was higher than that of *guaD2* under most conditions, with *guaD2* higher than *guaD1* under the remaining conditions (Fig 4B). Transcriptional regulation indicated that *P. aeruginosa* might coordinate the expression of two copies to adapt to the external environment, implying the transcriptional complementarity in response to external stimuli.

BCAAs belong to important nutrients in bacterial physiology, which are involved in supporting protein synthesis, intracellular signaling, regulating adaptation to amino acid starvation, and inducing the expression of virulence genes to promote proliferation and immunological escape [52]. Therefore, maintaining the intracellular concentration of BCAAs contributes to the survival of pathogenic bacteria in the host. Given the low contents of BCAAs in several host niches with μM level, including bronchoalveolar [53], blood [54] and nasal secretions [55], active transport of BCAAs is essential for supporting the pathogen lifestyle in clinical settings. Here, two transporters specific to BCAAs, BraB and BraZ, were widespread in *P. aeruginosa* strains, whereas only one copy, a homolog of BraB, was found in other *Pseudomonas* species (Fig 1 and S1 Fig), implying the possible importance of two copies in the adaptation of *P. aeruginosa* to the clinical settings. In fact, previous studies have found that the functional integrity of transporters specific to BCAAs is necessary for pathogenic bacteria to infect the host [56, 57]. Additional findings of two aspects supported that BraB and BraZ might contribute to the clinical adaptation of *P. aeruginosa*. Firstly, the two copies are complementary in terms of substrate specificity and coupling cations. BraB is specific for L-leucine and L-isoleucine and less specific for L-valine [58], whereas BraZ is specific for L-isoleucine and L-valine and less specific for L-leucine [59]. Despite both utilize the energy from the proton motive force to import BCAAs [52], the two copies differ from each other for coupling cations. BraB is a $Na^+$ or $Li^+$-coupled transport system [60, 61], and instead, BraZ is coupled with $H^+$ [59], implying the complementarity in their responses to the environmental acidity and alkalinity. Secondly, either BraB or BraZ is a transmembrane transporter, and the transcription

and translation of just one gene enables importing BCAAs [59, 62]. However, the transport system I nonspecific to BCAAs belongs to a transmembrane transport complex assembled by six proteins, with six genes enabling the import of BCAAs [63]. The energy consumption of BraB or BraZ is significantly lower than that of the transport system I. In addition, the transport system I functions mainly under adequate nutritional conditions, whereas BraB or BraZ works under conditions of limited external nutrition [52]. Given the low content of BCAAs in host-associated settings, the coexistence and functional complementary of BraB and BraZ might help *P. aeruginosa* survive in the host.

Guanine, guanine-derived nucleotides, and nucleosides are key biomolecules that maintain the normal life activities of bacteria [64, 65]. Here, we found that *P. aeruginosa* possessed two copies of GuaD, a key enzyme involved in guanine metabolism, and that GuaD1 and GuaD2 have diverged in evolutionary pathways and expression patterns. Although the functional difference of GuaD1 and GuaD2 for catalyzing guanine deamination has not been uncovered [48], their responses to different external conditions, or differences in expression regulation might further contribute to the adaptation to environmental stimuli. WGCNA identified 78 co-expressed genes of *guaD1*, as well as 77 co-expressed genes for *guaD2*. Within co-expressed genes of *guaD1*, the biological functions of 18 genes have been characterized (S7 Table). Importantly, PA1858 (*str*) turned out to be involved in resistance to multiple antibiotic treatments, and PA0847 (*diguanylate cyclase*), PA3319 (*plcN*), PA4988 (*waaA*) and PA5197 (*rimK*) could help *P. aeruginosa* to form the biofilms and infect hosts. For co-expressed genes of *guaD2*, PA1129 (*fosA*) could confer fosfomycin resistance to *P. aeruginosa* and PA0873 (*phhR*), PA1242 (*sprP*) and PA2273 (*soxR*) were involved in the formation of biofilms and infecting hosts (S8 Table). In summary, although there was no evidence to support that GuaD1 and GuaD2 could directly contribute to the adaptation of *P. aeruginosa* to the clinical environment, they might indirectly promote the response to clinical antibiotic treatment and to assist in infecting the host by coordinating the functions of their co-expressed genes.

As important nutrients in bacterial physiology, the intracellular availability of BCAAs directly affects the metabolic status of bacteria [52]. Both of two transporters specific to BCAAs in *P. aeruginosa*, BraB and BraZ, could regulate metabolic levels by altering the intracellular concentration of BCAAs and further promote the adaptation to external stimuli. In addition to the basic function of a gene, biological processes in which its co-expressed genes are involved may confer additional functions to it [66]. In this study, WGCNA identified 21 co-expressed genes of *braB*, as well as 29 co-expressed genes for *braZ* (Fig 4B). Within co-expressed genes of *braB*, *lpxO2* and *argC* turned out to be involved in resisting antibiotics (S9 Table). For co-expressed genes of *braZ*, seven of 12 genes with known functions have been found to help infect the host, and/or resist antibiotic treatments (S10 Table), including *dsbH*, *truA*, *trkH*, *htrB2*, *ampG*, *glpT* and *rarD*. Taken together, the coordination of *braB* and *braZ*, as well as respective co-expressed genes, helps *P. aeruginosa* adapt to clinical settings and infect the host.

The genes with two copies have also been found in other clinical pathogenic bacteria, such as *Staphylococcus aureus* [67] and *Escherichia coli* [68]. Qi et al found that two yoeB homologs might contribute to *S. aureus* planktonic growth, extracellular dependent biofilm formation, antibiotic tolerance, and virulence [67]. Two functional copies of the *irmA* gene were discovered in the enteroaggregative *E. coli* strain 042, and the *irmA_2244* allele appears to play a backup role to ensure IrmA expression when the *irmA_4509* allele loses its function, supporting their functional complementarity [68]. In addition to two substance metabolism genes in this study, a recent study also identified the presence of energy metabolism gene with two copies, glycerol kinase (glpK), in the *P. aeruginosa* population [69]. Tang et al found significantly different catalytic activity between GlpK1 and GlpK2, and identified the critical amino acid

Q70 involved in catalytic activity of GlpK by point mutation [69]. Taken together, the genes with two copies appear to enhance the environmental adaptability of pathogenic bacteria through functional differences and complementarity between two copies.

This study provides potential evidence that copy number increase and subsequent functional divergence of metabolic genes possibly not only enhance *P. aeruginosa*'s metabolic versatility, but also contribute to its adaptation in clinical settings. Whereas, there are still several limitations that exist. Firstly, this study lacks rigorous experimental validation to demonstrate that strains with two copies are indeed more adaptable in clinical settings than those with one copy. Future experimental work is needed. Secondly, although this study showed the potential of metabolic versatility via copy number increase of metabolic genes, whether the targeted intervention of a copy's function is feasible and effective still requires a lot of work. Finally, this study provides a paradigm to explore the evolution of metabolic genes at the population level, but there is still a huge gap between the results of this study and clinical applications.

## Conclusions

Two copies are widespread in *P. aeruginosa*, both for guanine deaminase and transporters specific to BCAAs. And GuaD1 and BraB were inherited from the ancestor of *Pseudomonas*, and GuaD2 was obtained via the HGT event, as well as about 9 kb region flanking *braZ* was acquired via evolutionary events, in the ancestors of *P. aeruginosa*. Functional divergence of GuaD1 and GuaD2, as well as BraB and BraZ, was supported by dN/dS distributions, transcriptional profiles, co-expressed genes and their GO terms. The coordination of two copies and their co-expressed genes, might help *P. aeruginosa* adapt to clinical settings.

## Supporting information

**S1 Fig. The copy number of BraT and GuaD in 391 *P. aeruginosa* strains.** The phylogeny of 391 *P. aeruginosa* strains was built using IQ-TREE based on the concatenation of the 120 marker proteins with 1000 bootstrap replicates. The scale bar represents 0.1 substitutions per site and the bootstrap values of major clades are shown at the nodes. The copy numbers of guanine deaminase (GuaD) or transporters specific to BCAA (BraT) are indicated by the inner strips, with white for two copies and grey for one copy. The isolation sources are indicated by the outer strips. Purple, brown, green, orange and light blue strips represent clinical-associated, animal-associated, plant-associated, soil-associated and other samples, respectively. Four representative strains are marked with black dots, including PAO1, LESB58, UCBPP_PA14 and PA7.
(TIF)

**S2 Fig. Comparative analysis of genomic regions flanking *braZ* of 13 strains with two *braZ* in *P. aeruginosa* referred to PAO1.** Colorful arrows and dark shading indicate the gene direction and nucleotide sequence identity of conserved regions. *braZ*_copy 1 in the sub-clade 1 of Fig 1A (right) shared the same genetic organization of genomic regions flanking *braB*, but with low identity of 59% between *braZ*_copy 1 and *braB* (left). *braZ*_copy 2 in the sub-clade 2 of Fig 1A (right) shared the same genetic organization of genomic regions flanking *braZ*, with high identity of 99% between *braZ*_copy 2 and *braZ* (right). The nucleotide sequence identity less than 90% are indicated. The scale bar indicates the length of 2.5kb nucleotides.
(TIF)

**S3 Fig. Comparative analysis of genomic regions flanking *braZ* and *braB* of three strains with one BraZ in *P. aeruginosa* referred to PAO1.** Colorful arrows and dark shading indicate the gene direction and nucleotide sequence identity of conserved regions. (**A**) The open

reading frame was frameshifted at the position 1041, and stop codon was found at the position 1067–1069 in *P. aeruginosa* PAK GCF_902172305.2 and *P. aeruginosa* PAK GCF_000568855.2, resulting in the deleted protein sequences of *braB*. (**B**) For *P. aeruginosa* PA1R GCF_000496645.1, *braZ*_copy in the sub-clade 1 of Fig 1A (right) sharing the same genetic organization of genomic regions flanking *braB*, but with low identity of 59% between *braZ*_copy and *braB*. About 80kb of genomic region flanking the original position of *braZ* was lost.
(TIF)

**S4 Fig. Comparative analysis of genomic regions flanking *braB* and *braZ* of six strains with one BraB in *P. aeruginosa* referred to PAO1.** Colorful arrows and dark shading indicate the gene direction and nucleotide sequence identity of conserved regions. (**A**) All six strains shared the same genetic organization of genomic regions flanking *braB*. (**B**) The open frame was frameshifted at the position 115, and stop codon was found at the position 142–144 in *P. aeruginosa* GCF_001516205.2 and *P. aeruginosa* GCF_002205355.1, resulting in the deleted protein sequences of *braZ*. For *P. aeruginosa* GCF_019434155.1 and *P. aeruginosa* GCF_019434175.1, about 41kb genomic region was lost in each strain. And the deleted genomic region ranges from 2116000 to 2157701 referred to *P. aeruginosa* PAO1, consisting of *braZ* with 2151755–2153068. (**C**) For *P. aeruginosa* GCF_001516365.2 and *P. aeruginosa* GCF_003288355.1, about 81kb genomic region was lost in each strain. And the deleted genomic region ranges from 2120000 to 2201000 referred to *P. aeruginosa* PAO1, consisting of *braZ* with 2151755–2153068.
(TIF)

**S5 Fig. Comparative analysis between genomic regions flanking *guaD* (A) or *braT* (B) in four representative strains of *P. aeruginosa* referred to PAO1.** Colorful arrows and dark shading indicate the gene direction and nucleotide sequence identity of conserved regions. No dark shading was found between genomic regions flanking two copies, except between *braB* and *braZ*, as well as *guaD1* and *guaD2*. The nucleotide sequence identity less than 90% are indicated. The scale bar indicates the length of 2.5kb nucleotides.
(TIF)

**S6 Fig. Two types of genomic regions flanking *guaD1* in *P. aeruginosa*.** (**A**) Pangenome graph of 391 *P. aeruginosa* genomes. The frame shows a zoom of genomic region containing *guaD1* and its flanking genes. (**B**) Two typical strains with one of two types of genomic regions flanking guaD1 in *P. aeruginosa*, respectively. The amplified zoom (left) and comparative analysis of genomic regions (right). The gene arrangement was indicated with identical colors.
(TIF)

**S7 Fig. Comparative analysis of genomic regions flanking *braT* in 10 species of *Pseudomonas* referred to PAO1.** Colorful arrows and dark shading indicate the gene direction and nucleotide sequence identity of conserved regions. The nucleotide sequence identity less than 80% are indicated. The scale bar indicates the length of 2.5kb nucleotides. There was no co-linearity between genomic regions flanking transporters specific to BCAAs in each species and that of *braZ*. (**A**) It is about four species in Clade I in Fig 2A (right) with their transporters specific to BCAAs more similar to *braB* than *braZ*. (**B**) It is about six species in Clade II in in Fig 2A (right) with their transporters specific to BCAAs more similar to *braZ* than *braB*.
(TIF)

**S8 Fig. Blasting against 10 species of *Pseudomonas* using about 101 kb of genomic region flanking *braB* (A) or *braZ* (B) in *P. aeruginosa* PAO1.** The 10 species refer to those species

used for comparative analysis with PAO1 in Fig 2C. The blue filled boxes on the line represent the length and the position of matched regions in each species relative to the query in PAO1 under a given covering depth. The proportion of matched regions occupying the query sequence is estimated and indicated with coverage referred to the query. About 9kb of genomic region flanking *braZ* was lost in other species, while 20kb of genomic region flanking *braB* maintained stable in other species in *Pseudomonas*.
(TIF)

**S9 Fig. The species tree (outside) and the phylogenetic tree of BraT (inside) of 386 γ-proteobacteria species.** The scale bar represents 0.5 substitution per site for the species tree, as well as 1 substitution per site for the protein tree. Species adjacent to *P. aeruginosa* BraB or BraZ are displayed in a zoom, with (a) for Clade I and (b) for Clade II.
(TIF)

**S10 Fig. Phylogenetic relationships of BraT in bacteria.** The scale bar indicates 1 substitution per site. **(A)** All hits in NR database were clustered by CID-HIT with 90% sequence identity as a threshold. For BarT in *Pseudomonas*, Clade I and Clade II were marked with light red and light blue consisting of BraB and BraZ in *P. aeruginosa*, respectively. The clades marked by grey boxes consist of the phylogenetic positions of BraB and BraZ, and their adjacent clades. **(B)** Phylogenetic tree of reconstructing the phylogenetic relationship using IQ-TREE for the protein sequences in the branch marked by the grey box in A. The representative protein sequences selected by CID-HIT were aligned with mafft and trimmed by trimAl. The resulting sequences were subject to the phylogenetic analysis using IQ-TREE. The possible evolutionary source (*Keratersia gyiorum*) of BraZ is highlighted with mulberry. **(C)** Comparative analysis of genomic region flanking *braT* in *K. gyiorum* and that of *braZ* in *P. aeruginosa* PAO1.
(TIF)

**S11 Fig. Density of candidate co-expressed genes of two copies about two metabolic genes in 64000 combinations.** Density stands for the distribution of co-expressed genes for a given number of combinations (Frequency). Normal distribution curves were added to determine μ and σ using the ggplot2 package. Based on the 2σ principle, the value of μ+2σ was used as the threshold to identify the most significantly co-expressed genes of *guaD1* and *guaD2*. Based on the 3σ principle, the value of μ+3σ was used as the threshold to identify the most significantly co-expressed genes *braB* and *braZ*.
(TIF)

**S12 Fig. Clustering analysis of expression profiles of co-expressed genes under 40 treatment conditions.** Z-scores by RMA function were used as the input. **(A)** Respective co-expressed genes of *guaD1* and *guaD2* were marked with the same colors referred to *guaD1* and *guaD2* in Fig 1A (center), respectively. **(B)** Respective co-expressed genes of *braB* and *braZ* were marked with the same colors referred to *braB* and *braZ* in Fig 1A (right), respectively.
(TIF)

**S1 Table. Metadata for selected 1945 strains in γ-proteobacteria.**
(XLSX)

**S2 Table. Metadata for habitat environment of 391 *P. aeruginosa* strains.**
(XLSX)

**S3 Table. Metadata for transcriptional datasets of *P. aeruginosa* strains.**
(XLSX)

**S4 Table. Number of guanine deaminases in each of selected 1945 strains in γ-proteobacteria.**
(XLSX)

**S5 Table. Number of transporters specific to BCAAs in each of selected 1945 strains in γ-proteobacteria.**
(XLSX)

**S6 Table. Frequency of co-expressed genes about two copies of two metabolic genes in 391 *P. aeruginosa* strains.**
(XLSX)

**S7 Table. Functions of co-expressed genes of *guaD1* in *P. aeruginosa* PAO1.**
(XLSX)

**S8 Table. Functions of co-expressed genes of *gauD2* in *P. aeruginosa* PAO1.**
(XLSX)

**S9 Table. Functions of co-expressed genes of *braB* in *P. aeruginosa* PAO1.**
(XLSX)

**S10 Table. Functions of co-expressed genes of *braZ* in *P. aeruginosa* PAO1.**
(XLSX)

## Author Contributions

**Formal analysis:** Yuqi Shi.

**Methodology:** Yuqi Shi.

**Software:** Yuqi Shi.

**Supervision:** Xiaohui Liang.

**Visualization:** Yutong Wu.

**Writing – original draft:** Yutong Wu.

**Writing – review & editing:** Xiaohui Liang.

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
