## [Decision Letter · Decision Letter 0]

30 Oct 2024

PONE-D-24-33220Evolution of two metabolic genes involved in nucleotide and amino acid metabolism in Pseudomonas aeruginosaPLOS ONE

Dear Dr. Liang,

Thank you for submitting your manuscript to PLOS ONE. After careful consideration, we feel that it has merit but does not fully meet PLOS ONE’s publication criteria as it currently stands. Therefore, we invite you to submit a revised version of the manuscript that addresses the points raised during the review process.

We look forward to receiving your revised manuscript.

Kind regards,

Rajesh P. Shastry, Ph.D

Academic Editor

PLOS ONE

Journal Requirements:

Additional Editor Comments (if provided):

Reviewers' comments:

Reviewer's Responses to Questions

**Comments to the Author**

1. Is the manuscript technically sound, and do the data support the conclusions?

Reviewer #1: Yes

Reviewer #2: Yes

2. Has the statistical analysis been performed appropriately and rigorously? 

Reviewer #1: Yes

Reviewer #2: Yes

3. Have the authors made all data underlying the findings in their manuscript fully available?

Reviewer #1: Yes

Reviewer #2: Yes

4. Is the manuscript presented in an intelligible fashion and written in standard English?

Reviewer #1: Yes

Reviewer #2: Yes

5. Review Comments to the Author

Reviewer #1: This research article from Wu, Shi, and Liang presents an interesting look into how a guanidine deaminase and a branch chain amino acid transporter evolved in the priority pathogen Pseudomonas aeruginosa. By understanding how P. aeruginosa and in turn other bacteria, we can gain an understanding of how the bacterium maintains homeostasis of the nucleotide pool within the cell. This article presents a comprehensive analysis on the evolution and acquisition of these important genes.

The discussion and conclusions within this research are impeccable, I found myself writing down questions while reading through the results which were subsequently comprehensively answered within the discussion. I would like to thank the authors for their attention to detail within this manuscript, it was enjoyable to read.

I have some questions and very minor corrections I believe are critical to evaluate and attempt to address to improve this manuscript. These are presented in the attached documentation.

Reviewer #2: 1. The manuscript relies heavily on correlative data without providing strong experimental evidence to support the conclusions drawn. For example, the functional roles of the co-expressed genes related to infection and antibiotic resistance are not adequately substantiated. More rigorous experimental validation is necessary to strengthen the claims made.

2. The interpretation of the dN/dS ratios and their implications for functional divergence is not sufficiently rigorous. The authors should provide a more critical analysis of how their findings relate to the broader context of evolutionary biology in P. aeruginosa.

3. The discussion does not adequately address the limitations of the study or the potential implications of the findings.

4. In the discussion authors should provide a stronger rationale for why these findings significantly advance our understanding of P. aeruginosa's metabolic versatility compared to previous studies.

5. The two hypotheses presented regarding the evolutionary source of BraZ are not adequately substantiated. While the authors provide some evidence for both, the discussion tends to be speculative, please justify.

6. The manuscript does not sufficiently engage with recent literature that could provide additional context for the findings.

6. PLOS authors have the option to publish the peer review history of their article (what does this mean?). If published, this will include your full peer review and any attached files.

Reviewer #1: **Yes: **Dr Samuel Wardell

Reviewer #2: No

---

## [Author Response · Author response to Decision Letter 0]

14 Nov 2024

Xiaohui Liang, Ph. D.

Department of Critical Care Medicine, Nanjing Drum Tower Hospital

Nanjing 210008, P. R. of China

Cell phone: +86 19852837758

439397013@qq.com

November 1, 2024

Dear Editor,

We thank you very much for giving us an opportunity to revise our manuscript and we appreciate you and reviewers very much for your comments and suggestions on our manuscript entitled “Evolution of two metabolic genes involved in nucleotide and amino acid metabolism in Pseudomonas aeruginosa” (PONE-D-24-33220). We have carefully read the reviews' comments and tried our best to revise our manuscript according to the reviewer's questions and suggestions. The detailed explanation of revision in response to the reviewers' concerns is given point by point in the following pages, and the part we have modified were marked in red in the revised manuscript. We greatly appreciate your encouragement, and thank you for consideration. Looking forward to hearing from you.

Sincerely yours,

Xiaohui Liang

Reviewer 1

Questions:

Can the authors comment on the possibility of a gene duplication events occurring in P. aeruginosa ancestors to acquire new genes which can then diversified and co-evolved, compared acquisition of these genes from external means. Is it more or less likely to occur via external ancestors like shown here?

Answers:

Thank you for your questions. It is a very interesting suggestion. According to our literature research, gene duplication events are common in organisms with cellular structures, including animals, plants, and microorganisms (Kuzmin et al., 2022). Previous studies have also found gene duplication-amplification (GDA) event or gene duplication event in P. aeruginosa. For example, Toussaint et al (2017) identified a gene duplication-amplification (GDA) event covering several genes, including the quorum-regulated nucleoside hydrolase gene, nuh, and PA0148, encoding an adenine deaminase in P. aeruginosa. And Gorecki et al (2020) revealed an ancient gene duplication as the origin of the MdtABC efflux pump in P. aeruginosa using phylogenetic analysis, and conformed that this gene duplication was an ancient event which occurred before the split of Proteobacteria into Alpha-, Beta- and Gamma- classes. Therefore, gene duplication events occurring in P. aeruginosa ancestors are possible. 

The main sources of new genes are gene replication events or horizontal gene transfer events. As the reviewer 1 pointed out that new genes could diversified or co-evolved, both for these two types of evolutionary events. Based on the presence patterns of new genes in related species, the evolutionary origin of new genes can be clarified. It will be an interesting evolutionary research to compare new genes obtained from gene replication with those obtained from external means. Whereas, this work aimed at revealing the evolutionary scenario of metabolic genes with two copies unique to P. aeruginosa, possibly via the horizontal gene transfer events.

Minor revisions:

General comment/ Please ensure throughout the nomenclature for genes and proteins is correct and consistent. For example, guaD (gene, italics) and GuaD (protein, non-italics capitalised first letter).

Answers:

Thank you for your questions and sorry for our unscientific writing about the nomenclature for genes and proteins. Here, we have carefully checked the manuscript and corrected the writing errors to ensure throughout the nomenclature for genes and proteins is correct and consistent in the revised manuscript.

General comment/ I find the introduction is lacking somewhat. More information could be included about what is currently known about the function of guanidine deaminase and BCAA specific transporters, and why they would be important to P. aeruginosa metabolism and evolution. (i.e., What do they do and why?)

Answers:

Thank you for your suggestion and sorry for lacking the detailed descriptions about the function of guanidine deaminase and BCAA specific transporters. Here, we have carefully added more information about what do they and why they would be important to P. aeruginosa metabolism and evolution in the revised manuscript (line74-81, page 4).

Ln 32/ “Little is known about the evolution of these two metabolic genes”. I understand you are referring to guaD and the BCAA transporter, however you only give the name of guaD. Please rephrase to refer to proteins or include the BCAA transporter gene name.

Answers:

Thank you for your suggestion and sorry for the unclear descriptions. Here, we have carefully corrected the corresponding descriptions in a clearer form (line32-33, page 2). About the BCAA transporter gene name, we did not discover one scientific name based on our literature research. Notably, the names of two BCAA transporters, BraB and BraZ, were unlike with that of guanine deaminase (guaD). Based on BraB and BraZ, we named the transporter specific to BCAAs as BraT by combining the transport characteristic with the previous names (Bra) for ease of reading and understanding in the revised manuscript (line31, page 2).

Ln 57/ Citation 14 is incorrect, this is not the primary source for the WHO’s classification of carbapenem resistant P. aeruginosa as a priority pathogen. Please update with the official citation directly from the WHO.

Answers:

Thank you for your question and sorry for the unscientific citation. Here, we have carefully checked and corrected the corresponding citation in the revised manuscript (line533-537, page 24).

Ln 60/ “has presented key challenges for this pathogen to adapt to environmental shifts including clinical antibiotic therapy and immune stress”

Answers:

Thank you for your suggestion and sorry for the unclear descriptions. Here, we have carefully corrected the corresponding descriptions in a clearer form according to your comments in the revised manuscript (line61, page 3).

Ln89-99/ I appreciate the authors having included specific version of the databases and software utilized, could you please include the citations for Pfam and TIGRFAM, and the version for MUSCLE, Faops, trimA1, IQ-TREE, and iTOL (if they exists)

Answers:

Thank you for your question and sorry for missing the specific versions of the databases and software utilized. Here, we have carefully checked and added the corresponding descriptions in the revised manuscript (line104-106, page 5). Significantly, no specific version information of Faops was found.

Ln103/ Please cite PANTHER.

Answers:

Thank you for your suggestion. Here, we have carefully checked and added the corresponding descriptions in the revised manuscript (line111, 115, page 6). 

Ln110/ I trust the BLASTP was used from BLAST+ via command line. Could you please provide version and citation for this.

Answers:

Thank you for your suggestion. The BLASTP was indeed used from BLAST+ via command line. Here, we have carefully checked and added the corresponding descriptions in the revised manuscript (line118, page 6). 

L111/ I am unsure what this sentence is trying to say.

Thank you for your suggestion. The description of “Stop analyzing when the number of all hits for each round remains the same.” was indeed redundant. We have carefully deleted the corresponding descriptions in the revised manuscript (line119-120, page 6).

Ln118/ Please provide a version for Clinker if it exists.

Answers:

Thank you for your question. Significantly, no specific version information of Clinker was found in the corresponding reference.

Ln120/ As above, for PPanGGOLiN and Gephi.

Thank you for your question and sorry for missing the specific versions of the software utilized. Here, we have carefully checked and added the corresponding descriptions in the revised manuscript (line129-130, page 6). 

Ln126/ Please include ggplot2 version used.

Thank you for your question and sorry for missing the specific versions of the software utilized. Here, we have carefully checked and added the corresponding descriptions in the revised manuscript (line136, page 7). 

Ln131/ likewise with affy.

Thank you for your question. Significantly, no specific version information of affy was found in the corresponding reference.

Ln140/ and with eggNOG.

Thank you for your question and sorry for missing the specific versions of the software utilized. Here, we have carefully checked and added the corresponding descriptions in the revised manuscript (line150, page 7). 

L141/ Include version and citation for clusterProfiler.

Thank you for your question and sorry for missing the specific versions of the software utilized. Here, we have carefully checked and added the corresponding descriptions in the revised manuscript (line152, page 7). 

Ln142/ Was this a corrected p value or raw p value as clusterProfiler can provide both.

Thank you for your question and sorry for the unclear descriptions. Here, we have carefully checked and corrected the corresponding descriptions in the revised manuscript (line152, page 7). 

Ln150/ I believe alongside Sood et al, this would be a more appropriate citation (10.3389/fmicb.2015.01036).

Thank you for your suggestion. We have carefully checked and changed the reference in the revised manuscript according to your comments (line161, page 8).

Ln164/ Is there a mislabelling in Supplementary Figure 1, I cannot distinguish the two copies of BraZ, Additionally, in these supplementary figures, genes and loci are provided but when referring to bra, or gua, you label them with protein nomenclature, see ‘general comment’ above.

Thank you for your question and sorry for the unclear descriptions. Here, we have carefully checked and corrected the corresponding descriptions in the Supplemental Figures. And we carefully corrected the writing errors to ensure throughout the nomenclature for genes and proteins is correct and consistent in the revised manuscript. 

Ln165/ “Genomic analysis of four commonly used reference isolates, namely …”.

Thank you for your suggestion. We have carefully checked and changed the corresponding descriptions in the revised manuscript according to your comments (line177, page 9).

Ln167/ should this read “flanking the two copies of the two metabolic genes, except…”.

Thank you for your suggestion. We have carefully checked and changed the corresponding descriptions in the revised manuscript according to your comments (line179, page 9).

Ln 175/ is it possible to include gene identification for the 5 flanking genes either side in figure 1C like shown in Supplementary figure 6B. I believe this will enhance clarity.

Thank you for your suggestion. We have carefully checked and added the gene identification for the 5 flanking genes either side in figure 1C in the revised manuscript according to your comments.

Ln243/ Version of CD-HIT.

Thank you for your question and sorry for missing the specific versions of the software utilized. Here, we have carefully checked and added the corresponding descriptions in the revised manuscript (line256, page 12).

Ln243-250/ This is borderline methods, perhaps this could be included in methods rather than in results section. Although I understand why the authors have done it this way.

Thank you for your suggestion. As you pointed out, those corresponding descriptions are indeed borderline methods. Whereas, we tried transferring the corresponding descriptions in results section to the methods section according to your suggestion. Unfortunately, we cannot describe the corresponding analysis content concisely and logically due to the condition and logicality of subsequent analysis process. Therefore, we still suggest that the borderline methods should be included in results section for better understanding by readers.

References:

Kuzmin E, Taylor JS, Boone C. Retention of duplicated genes in evolution. Trends Genet. 2022,8(1):59-72. doi: 10.1016/j.tig.2021.06.016.

Toussaint JP, Farrell-Sherman A, Feldman TP, Smalley NE, Schaefer AL, Greenberg EP, Dandekar AA. Gene duplication in Pseudomonas aeruginosa improves growth on adenosine. J Bacteriol. 2017, 199(21):e00261-17. doi: 10.1128/JB.00261-17. 

Górecki K, McEvoy MM. Phylogenetic analysis reveals an ancient gene duplication as the origin of the MdtABC efflux pump. PLoS One. 2020, 15(2):e0228877. doi: 10.1371/journal.pone.0228877.

Reviewer 2

Questions:

1. The manuscript relies heavily on correlative data without providing strong experimental evidence to support the conclusions drawn. For example, the functional roles of the co-expressed genes related to infection and antibiotic resistance are not adequately substantiated. More rigorous experimental validation is necessary to strengthen the claims made.

Answers:

Thank you for your questions. 

Based on literature research, it has encountered bottlenecks to explore novel antibiotics from the perspective of antibiotic resistance genes or to discover alternative treatment strategies for drug-resistant bacterial infections. In recent years, several high-quality studies on conferring antibiotic resistance to bacteria via metabolic changes have been published (Lopatkin et al., 2021; Palomino et al., 2023). These works have provided us with new cut-in points for dealing with antibiotic resistance. Focusing on the evolution of metabolism-related genes, an initial survey revealed that 14 metabolism-related genes exhibited two copies in several typical strains of Pseudomonas aeruginosa, whereas only one copy was found in other investigated species. This study focuses on the evolution of two genes involved in material metabolism including guanine deaminase and transporters specific to BCAAs, and aims to reveal the evolutionary characteristics of these two metabolism genes at the population level of P. aeruginosa through multi-omics analysis. As pointed out by you, this study relies heavily on correlative data without providing strong experimental evidence to support the conclusions drawn.

Firstly, the conclusion drawn from this study is mainly to illustrate the possible or speculative relationship between the existence of metabolic diversity and clinical adaptability. Secondly, we do not have the capability to directly conduct control experiments on highly pathogenic bacteria such as P. aeruginosa to demonstrate that the strains with two copies do indeed have stronger environmental adaptability than those strains with a single copy. As to transporters specific to BCAAs, functional differences of BraB and BraZ in P. aeruginosa have been proven by previous experimental evidence (Hoshino 1979; Hoshino and Kageyama 1979; Uratani et al., 1989; Hoshino et al., 1990; Hoshino et al., 1991). Given this, the presence of two copies can indeed enrich the metabolic diversity of P. aeruginosa, which may help it cope with environmental stress. Finally, the co-expressed genes identified at the transcriptional level were only used to illustrate different co-expression networks between the two copies, indirectly supporting the possibility that the presence of two copies may increase the metabolic diversity of P. aeruginosa. It is not the focus of this study that more rigorous experimental validation is conducted to strengthen the claims made.

2. The interpretation of the dN/dS ratios and their implications for functional divergence is not sufficiently rigorous. The authors should provide a more critical analysis of how their findings relate to the broader context of evolutionary biology in P. aeruginosa.

Thank you for your suggestion. Firstly, this study infers the divergence of evolutionary paths between two copies based on the differences in distribution patterns of dN/dS ratios (line272-276, page 13). Secondly, this study provides possible inferences of functional divergence of two copies through evidence from multiple perspectives, including distribution patterns of dN/dS ratios, transcriptional profiles, co-expressed genes and their GO terms. Finally, as to two copies of transporters specific to BCAAs, functional differences of BraB and BraZ in P. aeruginosa have been proven by previous experimental evidence (Hoshino 1979; Hoshino and Kageyama 1979; Uratani et al., 1989; Hoshino et al., 1990; Hoshino et al., 1991). Although experimental studies on the two copies of guanine deaminase have not been reported, the results obtained from multiple perspectives in this study support th

---

## [Decision Letter · Decision Letter 1]

4 Dec 2024

Evolution of two metabolic genes involved in nucleotide and amino acid metabolism in Pseudomonas aeruginosa

PONE-D-24-33220R1

Dear Dr. Liang,

We’re pleased to inform you that your manuscript has been judged scientifically suitable for publication and will be formally accepted for publication once it meets all outstanding technical requirements.

Kind regards,

Rajesh P. Shastry, Ph.D

Academic Editor

PLOS ONE

Additional Editor Comments (optional):

Reviewers' comments:

Reviewer's Responses to Questions

**Comments to the Author**

1. If the authors have adequately addressed your comments raised in a previous round of review and you feel that this manuscript is now acceptable for publication, you may indicate that here to bypass the “Comments to the Author” section, enter your conflict of interest statement in the “Confidential to Editor” section, and submit your "Accept" recommendation.

Reviewer #1: All comments have been addressed

Reviewer #2: All comments have been addressed

2. Is the manuscript technically sound, and do the data support the conclusions?

Reviewer #1: Yes

Reviewer #2: Yes

3. Has the statistical analysis been performed appropriately and rigorously? 

Reviewer #1: Yes

Reviewer #2: Yes

4. Have the authors made all data underlying the findings in their manuscript fully available?

Reviewer #1: Yes

Reviewer #2: Yes

5. Is the manuscript presented in an intelligible fashion and written in standard English?

Reviewer #1: Yes

Reviewer #2: Yes

6. Review Comments to the Author

Reviewer #1: I appreciate the authors attention to detail in addressing the comments from the reviewers, I have no additional comments to add

Reviewer #2: Overall, it is a clear and well defined manuscript. all the queries are addressed well and included in the revised manuscript.

7. PLOS authors have the option to publish the peer review history of their article (what does this mean?). If published, this will include your full peer review and any attached files.

Reviewer #1: No

Reviewer #2: No

---

## [Editor Report · Acceptance letter]

5 Dec 2024

PONE-D-24-33220R1 

PLOS ONE

Dear Dr. Liang, 

I'm pleased to inform you that your manuscript has been deemed suitable for publication in PLOS ONE. Congratulations! Your manuscript is now being handed over to our production team.

Kind regards, 

on behalf of

Dr. Rajesh P. Shastry 

Academic Editor

PLOS ONE